# On the True Number of COVID-19 Infections: Effect of Sensitivity, Specificity and Number of Tests on Prevalence Ratio Estimation

**DOI:** 10.3390/ijerph17155328

**Published:** 2020-07-24

**Authors:** Eitan Altman, Izza Mounir, Fatim-Zahra Najid, Samir M. Perlaza

**Affiliations:** 1INRIA, Centre de Recherche de Sophia Antipolis-Mediterranee, 2004 Route des Lucioles, BP 93, 06902 Sophia Antipolis CEDEX, France; eitan.altman@inria.fr; 2Laboratoire Informatique d’Avignon, Campus Jean-Henri Fabre, Avignon Universite, 84 911 Avignon, France; 3Laboratory of Information, Network and Communication Sciences, Avignon Universite, 75013 Paris, France; 4Centre Hospitalier Universitaire de Nice, School of Medicine, Université Côte D’Azur, 30 Voie Romaine, 06000 Nice, France; mounir.i@chu-nice.fr; 5Centre Hospitalier Universitaire Amiens Picardie, School of Medicine, Université de Picardie Jules Verne, 1 Rue du Professeur Christian Cabrol, 80054 Amiens, France; najid.fatim-zahra@chu-amiens.fr

**Keywords:** SARS-CoV-2, Covid-19, cross-sectional studies, prevalence ratio, sensitivity and specificity, molecular, serological and medical imaging diagnostics, number of infections, false positive and false negative probabilities, policy-making and testing campaigns

## Abstract

In this paper, a formula for estimating the prevalence ratio of a disease in a population that is tested with imperfect tests is given. The formula is in terms of the fraction of positive test results and test parameters, i.e., probability of true positives (sensitivity) and the probability of true negatives (specificity). The motivation of this work arises in the context of the COVID-19 pandemic in which estimating the number of infected individuals depends on the sensitivity and specificity of the tests. In this context, it is shown that approximating the prevalence ratio by the ratio between the number of positive tests and the total number of tested individuals leads to dramatically high estimation errors, and thus, unadapted public health policies. The relevance of estimating the prevalence ratio using the formula presented in this work is that precision increases with the number of tests. Two conclusions are drawn from this work. First, in order to ensure that a reliable estimation is achieved with a finite number of tests, testing campaigns must be implemented with tests for which the sum of the sensitivity and the specificity is sufficiently different than one. Second, the key parameter for reducing the estimation error is the number of tests. For a large number of tests, as long as the sum of the sensitivity and specificity is different than one, the exact values of these parameters have very little impact on the estimation error.

## 1. Introduction

In the absence of a vaccination or effective medical treatment against the SARS-CoV-2, the global population must cohabitate with the virus. For succeeding in this task, different strategies to slow down the outbreak can be implemented, for example, encouraging social distancing, isolation of infected individuals, mobility restrictions, lockdowns, and contact tracing. The main objective is to guarantee that the number of infected individuals that develop critical forms of symptoms does not exceed the capacity of local health care systems. Nonetheless, most of the strategies to slow down the outbreak induce dramatic economical consequences, and thus, public health policies must be designed based on reliable predictions of the evolution of the pandemic to minimize undesired effects on the global economy. For doing so, estimating the values of variables such as the proportion of susceptible, infected and recovered individuals in the population, among other variables, is of paramount importance. This is due to the fact that such variables are the inputs of mathematical models that help to predict the evolution of the pandemic [1,2], and thus, impact public health policy-making. Reliable estimations of these variables can be achieved in part by testing the population. Nonetheless, diagnosing SARS-CoV-2 is a challenging task given that designing highly reliable tests for massive testing is still an open research problem, c.f., [3,4,5].

In the general realm of epidemiology, the reliability of tests is measured in terms of two parameters: sensitivity and specificity. The former is the probability with which a test is able to correctly identify the presence of a condition, for example, a SARS-Cov-2 infection. Alternatively, the latter is the probability with which a test is able to correctly identify the absence of such condition. Within this context, the main contribution of this work is a mathematical formula for estimating the fraction of individuals that exhibit the condition in a population in which every individual has been tested once with identical unreliable tests. In the following, this fraction is referred to as the prevalence ratio [2]. In these terms, the main result is Theorem 1 in Section 4, which presents an estimator of the prevalence ratio in terms of the sensitivity, specificity and the fraction of positive test results. More importantly, the estimation error induced by such estimator is proved to decrease with the number of tests.

The novelty of this work with respect to existing methods for estimating the prevalence ratio, such as the method of multipliers, capture and recapture methods, among others [2,6], is that it takes into account the effects of both false positive and false negative probabilities. This consideration has already been discussed by several authors, c.f., [7,8,9,10]. Nonetheless, a simple general formula for estimating prevalence ratios in terms of the sensitivity, specificity, and the fraction of positive test results is not available in current literature. This said, the prevalence ratio estimation presented in this work is based exclusively on the results of data obtained through testing campaigns with unreliable binary tests. The main hypotheses adopted in this work are: (a) Individuals are tested once and test results are independent of each other; and (b) the prevalence ratio is assumed constant during the duration of the testing campaign. This breaks away from the studies based on mathematical regressions in which some assumptions on the proabability distribution of the random variables are adopted and whose correctness is often the ground of vivid discussions, c.f., [11,12,13,14].

The main conclusions of this work are:(i)The number of positive tests might be drastically different than the number of infected individuals in a population depending on the sensitivity and specificity of the tests. Hence, the ratio between the number of positive tests and the total number of tested individuals is not a reliable estimation of the prevalence ratio;(ii)Testing campaigns using tests for which the sum of the sensitivity and specificity is different than one, always allow a reliable estimation of the number of infected individuals when a sufficiently large number of individuals is tested in the population (Lemma 1 in Section 4);(iii)Testing campaigns using a test for which the sum of the sensitivity and the specificity is equal to one, lead to data from which it is impossible to estimate the prevalence ratio independently of the number of tested individuals (Lemma 7 in Section 4); and(iv)When the objective is to estimate the prevalence ratio in a population, the key parameter for reducing the estimation error is the number of tests (Lemma 5 in Section 4). That is, as long as the sum of the sensitivity and specificity is different than one, and a large number of test results is available, the exact values of both sensitivity and specificity have very little impact on the estimation error.

The remaining sections of this paper are organized as follows: Section 2 presents a brief overview of the tests for diagnosing SARS-CoV-2 and the reliability of the existing tests; Section 3 formulates the problem of estimating the prevalence ratio taking into account the sensitivity and specificity of the tests; Section 4 presents an estimator of the prevalence ratio using data obtained from unreliable tests, and the proofs of the main results; Section 5 introduces some examples in which the impact of the sensitivity, specificity and number of tests on the estimation error is numerically analyzed; Section 6 concludes this work.

## 2. Case Study: SARS-CoV-2

Tests for SARS-CoV-2 can be broadly divided into three groups: virological tests, serological tests, and tests based on medical imaging. Each of these groups provide information about different aspects of the infection and exhibit different reliability parameters.

### 2.1. Virological Tests

Virological tests inform about the presence of the SARS-CoV-2 virus genome in nasopharyngeal (nasal swab) or oropharyngeal swabs (oral swab), blood, anal swab, urine, stool, and sputum samples [15]. Individuals with positive virological tests are declared capable of contaminating others, and thus, virological tests are central in decision-making and policy-making, c.f. [3,5].

The reliability of virological tests in terms of sensitivity and specificity depends on a variety of parameters. These parameters include the type of clinical specimen, the materials and methods used for obtaining the specimens, specimen transportation, viral density of patients, and human errors in data processing in laboratories. In the case of respiratory specimens, viral density appears to play a central role in the sensitivity and specificity of virological tests, c.f., [16,17]. This stems from the fact that during the first week after infection, the virus can be detected by nasopharyngeal or oropharyngeal swabs. During the second week and later, the virus might disappear in the upper parts of the respiratory system and migrate to the bronchial tube and the lungs. From the studies in [16,17], it appears that specimens from the lower respiratory track increase the sensitivity and specificity of virological tests.

Virological tests are based on several techniques: (a) Reverse transcription polymerase chain reaction (RT-PCR), c.f., [18,19]; and (b) Reverse transcription loop-mediated isothermal amplification (RT-LAMP), c.f., [20,21]; and (c) other techniques, c.f., [19,22,23].

### 2.2. Serological Tests

Serological tests determine whether an individual has developed anti-bodies or antigens against the SARS-CoV-2 virus. Nonetheless, an individual produces anti-bodies against SARS-CoV-2 only several days after contracting the infection. Typically, the time between infection and the production of anti-bodies ranges from seven to fourteen days, c.f., [24,25,26]. Serological tests are based on the enzyme linked immunosorbent assay (ELISA) and exhibit high specificity and sensitivity, after fourteen days of infections [24]. This drastically limits the use of serological tests in the early detection of the infection and policy-making, c.f., [3,4]. In a nutshell, on the one hand, a serological test answers the question whether an individual is or has been infected. On the other hand, serological tests do not allow determining whether an individual has immunity to the SARS-CoV-2 virus or whether the individual is currently spreading the virus. Up to the day of publication of this paper, serological tests are not considered for massive testing in France, c.f., [4].

### 2.3. Medical Imaging

Medical Imaging for detection of SARS-CoV-2 includes chest X-Ray and chest computed tomography (CT) scans, which reveal ground-glass opacities and consolidations in the periphery of the lungs of infected individuals [27]. Nonetheless, the sensitivity and specificity of CT depends on the experience of radiologists to distinguish SARS-CoV-2 pneumonia from non-SARS-CoV-2 pneumonia [28]. In [29], it is reported that the sensitivity of CT is better than the one achieved by RT-PCR tests.

## 3. Prevalence Ratio and Unreliable Tests

Consider a population subset of *n* individuals whose state is either susceptible (*S*) or infected (*I*) and assume that all individuals of this population subset are tested with the same type of test. Let the actual state of such *n* individuals be represented by the vector $x≜(x_{1}$, $x_{2}$, *…*, $x_{n})$. That is, for all t∈{1,2,…,n}, it follows that $x_{t}\in \left\{I,S\right\}$ is the true state of the individual *t*. The result of testing individual *t* is denoted by $y_{t}\in \left\{I,S\right\}$. Hence, the outcome of a testing campaign over such population is a vector $y≜(y_{1}$, $y_{2}$, *…*, $y_{n})\in {\left\{I,S\right\}}^{n}$. Due to the fact that tests possess strictly positive probabilities of false negatives and false positives, the vectors x and y might be different. That is, some individuals that are infected could have been declared susceptible and vice versa.

A central observation in this analysis is that a test for determining whether an individual is contaminated by SARS-CoV-2 can be modeled by a random transformation $P_{Y|X}$ for which the input and output sets are {I,S}. More specifically, if an individual whose state is x∈{I,S} is tested, the result y∈Y is observed with probability $P_{Y|X}\left(y|x\right)$. Figure 1 shows this binary-input binary-output model.

Using this notation, the sensitivity of the test is $P_{Y|X}\left(I|I\right)$; and the specificity of the test is $P_{Y|X}\left(S|S\right)$. The probability of a false positive is $P_{Y|X}\left(I|S\right)=1-P_{Y|X}\left(S|S\right)$; and the probability of a false negative is $P_{Y|X}\left(S|I\right)=1-P_{Y|X}\left(I|I\right)$. This said, a test is fully described by any of the following pairs of parameters:The sensitivity and the specificity;The sensitivity and the probability of a false positive;The probability of a false negative and the specificity; orThe probability of a false negative and the probability of a false positive.

Let *X* be random variable taking values in {I,S} and denote by $P_{X}:\left\{I,S\right\}\to \left[0,1\right]$ its probability distribution such that $P_{X}\left(I\right)$ is the actual fraction of infected individuals among the *n* individuals. That is, $P_{X}\left(I\right)$ is the *prevalence ratio* of SARS-Cov-2 in this population subset. For this reason, the probability distribution $P_{X}$ is referred to as the *ground-truth input probability distribution.* Let *Y* be a second random variable taking values in {I,S} such that its joint probability distribution with *X* is $P_{XY}$ and for all $\left(x,y\right)\in {\left\{I,S\right\}}^{2}$,
(1)$$P_{XY}\left(x,y\right)=P_{X}\left(x\right)P_{Y|X}\left(y|x\right),$$
where the conditional distribution $P_{Y|X}$ is the test. See, for instance, Figure 1. Often, the probability distribution $P_{Y}$ is referred to as the *ground-truth output probability distribution* and it is obtained as the marginal of $P_{XY}$. That is, for all y∈{I,S},
(2)$$P_{Y}\left(y\right)=\sum_{x\in \{I,S\}}P_{X}\left(x\right)P_{Y|X}\left(y|x\right).$$

The problem consists in using the data y obtained through a testing campaign with tests in which parameters are modeled by $P_{Y|X}$ to determine the fraction $P_{X}\left(I\right)$ of infected individuals in the population, i.e., the prevalence ratio. More formally, the problem can be stated as follows: Consider two random variables *X* and *Y* with the joint probability distribution $P_{XY}$ in (Equation 1). The problem consists in estimating the probability distribution $P_{X}$ based only on *n* realizations $y_{1}$, $y_{2}$, *…*, $y_{n}$ of the random variable *Y*, with *n* a finite integer. This problem is reminiscent to the problem of *population recovery* introduced in [30] and further studied in [31,32].

## 4. Estimation of the Prevalence Ratio Using Unreliable Tests

Given the data $y\in {\left\{I,S\right\}}^{n}$ collected during a test campaign, the fraction of the population reporting positive and negative tests form an empirical distribution denoted by ${\bar{P}}_{Y}^{\left(n\right)}$ on the set {I,S} such that,
(3a)$$\begin{array}{ccc} {\bar{P}}_{Y}^{\left(n\right)}\left(I\right) & ≜ & \frac{1}{n}\sum_{t=1}^{n}{𝟙}_{\left\{I=y_{t}\right\}},\mathrm{and} \end{array}$$
(3b)$$\begin{array}{ccc} {\bar{P}}_{Y}^{\left(n\right)}\left(S\right) & ≜ & \frac{1}{n}\sum_{t=1}^{n}{𝟙}_{\left\{S=y_{t}\right\}}, \end{array}$$
where ${𝟙}_{\left\{\cdot \right\}}$ is the indicator function. Essentially, ${\bar{P}}_{Y}^{\left(n\right)}$ is a counting probability measure for which the values ${\bar{P}}_{Y}^{\left(n\right)}\left(I\right)$ and ${\bar{P}}_{Y}^{\left(n\right)}\left(S\right)$ represent the fraction of positive and negative test results. Hence, ${\bar{P}}_{Y}^{\left(n\right)}\left(I\right)+{\bar{P}}_{Y}^{\left(n\right)}\left(S\right)=1$. In the following, such probability measure is often referred to as the *output empirical distribution* obtained from the data y.

Let ${\hat{P}}_{X}^{\left(n\right)}:\left\{I,S\right\}\to R$ be a function representing the estimation of $P_{X}$ based on the data y. The error induced by estimating $P_{X}$ using ${\hat{P}}_{X}^{\left(n\right)}$ can be measured by the total variation, which is denoted by ${∥P_{X}-{\hat{P}}_{X}^{\left(n\right)}∥}_{\mathrm{TV}}$ and satisfies,
(4)$$\begin{array}{ccc} {∥P_{X}-{\hat{P}}_{X}^{\left(n\right)}∥}_{\mathrm{TV}} & ≜ & \frac{1}{2}\left(|P_{X}\left(I\right)-{\hat{P}}_{X}^{\left(n\right)}\left(I\right)\left|+\right|P_{X}\left(S\right)-{\hat{P}}_{X}^{\left(n\right)}\left(S\right)|\right) \end{array}$$
(5)$$\begin{array}{cc} = & |P_{X}\left(I\right)-{\hat{P}}_{X}^{\left(n\right)}\left(I\right)|. \end{array}$$

Note that in the case of binary tests, the total variation is simply the absolute difference between the actual prevalence ratio $P_{X}\left(I\right)$ and the estimate ${\hat{P}}_{X}^{\left(n\right)}\left(I\right)$.

### 4.1. Main Result

The following theorem presents the main result of this work.

**Theorem** **1.**
*Consider a population of n individuals whose true ratio of infected (I) and susceptible (S) individuals is $P_{X}\left(I\right)$ and $P_{X}\left(S\right)=1-P_{X}\left(S\right)$, respectively, with $P_{X}\left(I\right)\in \left[0,1\right]$. Assume that all individuals of such population are tested with a test $P_{Y|X}$ that satisfies*
(6)$$P_{Y|X}\left(S|S\right)+P_{Y|X}\left(I|I\right)\neq 1.$$

*Let ${\bar{P}}_{Y}^{\left(n\right)}$ be the resulting output empirical probability distribution in (3) and assume that ${\bar{P}}_{Y}^{\left(n\right)}\left(I\right)$ satisfies the following condition,*
(7)$$\min\left\{P_{Y|X}\left(I|I\right),P_{Y|X}\left(I|S\right)\right\}⩽{\bar{P}}_{Y}^{\left(n\right)}\left(I\right)⩽\max\left\{P_{Y|X}\left(I|I\right),P_{Y|X}\left(I|S\right)\right\}.$$

*Then, the estimator ${\hat{P}}_{X}^{\left(n\right)}:\left\{I,S\right\}\to R$ of $P_{X}$, such that*
(8a)$$\begin{array}{ccc} {\hat{P}}_{X}^{\left(n\right)}\left(I\right) & = & \frac{1-{\bar{P}}_{Y}^{\left(n\right)}\left(I\right)-P_{Y|X}\left(S|S\right)}{1-P_{Y|X}\left(I|I\right)-P_{Y|X}\left(S|S\right)},and \end{array}$$
(8b)$$\begin{array}{ccc} {\hat{P}}_{X}^{\left(n\right)}\left(S\right) & = & \frac{{\bar{P}}_{Y}^{\left(n\right)}\left(I\right)-P_{Y|X}\left(I|I\right)}{1-P_{Y|X}\left(I|I\right)-P_{Y|X}\left(S|S\right)}, \end{array}$$

*forms a probability measure that satisfies*
(9)$$\lim_{n\to \infty }{∥P_{X}-{\hat{P}}_{X}^{\left(n\right)}∥}_{\mathrm{TV}}=0,almost\;surely.$$


In a nutshell, Theorem 1 states that approximating the prevalence ratio $P_{X}$ by ${\hat{P}}_{X}^{\left(n\right)}$ induces an error that vanishes when the number of tests *n* increases. Nonetheless, despite the fact that ${\hat{P}}_{X}^{\left(n\right)}\left(I\right)+{\hat{P}}_{X}^{\left(n\right)}\left(S\right)=1$, it holds that for a small number of tests *n*, ${\hat{P}}_{X}^{\left(n\right)}\left(I\right)$ and ${\hat{P}}_{X}^{\left(n\right)}\left(S\right)$ do not necessarily form a probability measure. That is, it might be observed that either ${\hat{P}}_{X}^{\left(n\right)}\left(I\right)<0$ and ${\hat{P}}_{X}^{\left(n\right)}\left(S\right)>1$; or ${\hat{P}}_{X}^{\left(n\right)}\left(I\right)>1$ and ${\hat{P}}_{X}^{\left(n\right)}\left(S\right)<0$. Later, in Lemma 4, it is shown that with a large number of test results, the fraction of positive results ${\bar{P}}_{Y}^{\left(n\right)}\left(I\right)$ satisfies the inequalities in (Equation 7). Note also that the condition in (Equation 7) is necessary and sufficient to observe that $0⩽{\hat{P}}_{X}^{\left(n\right)}\left(I\right)⩽1$ in Theorem 1. This highlights the need for a sufficiently large number of tests in order to obtain a valid estimation of $P_{X}\left(I\right)$ using Theorem 1.

Finally, note that the formulas in (8) are given in terms of the sensitivity $P_{Y|X}\left(I|I\right)$ and specificity $P_{Y|X}\left(S|S\right)$ of the test. Nonetheless, it can be expressed in terms of the probabilities of a false positive and a false negative, or any combination of the parameters describing the test. The following corollary shows the formulas in (8) in terms of the probabilities of a false positive $P_{Y|X}\left(I|S\right)$ and a false negative $P_{Y|X}\left(S|I\right)$.

**Corollary** **1.**
*Consider a population of n individuals whose true ratio of infected (I) and susceptible (S) individuals is $P_{X}\left(I\right)$ and $P_{X}\left(S\right)=1-P_{X}\left(S\right)$, respectively, with $P_{X}\left(I\right)\in \left[0,1\right]$. Assume that all individuals of such population are tested with a test $P_{Y|X}$ that satisfies (Equation 6). Let ${\bar{P}}_{Y}^{\left(n\right)}$ be the resulting output empirical probability distribution in (3) and assume that ${\bar{P}}_{Y}^{\left(n\right)}\left(I\right)$ satisfies condition (Equation 7). Then, the estimator ${\hat{P}}_{X}^{\left(n\right)}:\left\{I,S\right\}\to R$ of $P_{X}$, such that*
(10a)$$\begin{array}{ccc} {\hat{P}}_{X}^{\left(n\right)}\left(I\right) & = & \frac{{\bar{P}}_{Y}^{\left(n\right)}\left(I\right)-P_{Y|X}\left(I|S\right)}{1-P_{Y|X}\left(S|I\right)-P_{Y|X}\left(I|S\right)},and \end{array}$$
(10b)$$\begin{array}{ccc} {\hat{P}}_{X}^{\left(n\right)}\left(I\right) & = & \frac{1-P_{Y|X}\left(S|I\right)-{\bar{P}}_{Y}^{\left(n\right)}\left(I\right)}{1-P_{Y|X}\left(S|I\right)-P_{Y|X}\left(I|S\right)}, \end{array}$$

*forms a probability measure that satisfies (Equation 9).*


### 4.2. Proof of Theorem  1

The proof of Theorem 1 leverages the following intuition: Under the assumption that ${\bar{P}}_{Y}^{\left(n\right)}$, which is obtained from the data y as in (3), is a valid estimation of the ground-truth output probability distribution $P_{Y}$, i.e., it satisfies (Equation 7), then a distribution ${\hat{P}}_{X}^{\left(n\right)}$ that satisfies
(11)$$\left(\begin{array}{c} {\bar{P}}_{Y}^{\left(n\right)}\left(I\right)\\ {\bar{P}}_{Y}^{\left(n\right)}\left(S\right) \end{array}\right)=\left(\begin{array}{cc} P_{Y|X}\left(I|I\right) & P_{Y|X}\left(I|S\right)\\ P_{Y|X}\left(S|I\right) & P_{Y|X}\left(S|S\right) \end{array}\right)\left(\begin{array}{c} {\hat{P}}_{X}^{\left(n\right)}\left(I\right)\\ {\hat{P}}_{X}^{\left(n\right)}\left(S\right) \end{array}\right),$$
is a good estimation of the input probability distribution $P_{X}$. This intuition builds upon the observation that the output distribution ${\bar{P}}_{Y}^{\left(n\right)}$ induced by the data, must be the marginal of a joint distribution consisting of the product of the conditional $P_{Y|X}$ and the input distribution. That is, for all y∈{I,S},
$${\bar{P}}_{Y}^{\left(n\right)}\left(y\right)=\sum_{x\in \{I,S\}}P_{Y|X}\left(y|x\right){\hat{P}}_{X}^{\left(n\right)}\left(x\right),$$
which is equivalent to the system in (Equation 11).

With this intuition in mind, the proof proceeds as follows. First, it is shown that under the condition in (Equation 6), there exists a unique pair $\left({\hat{P}}_{X}^{\left(n\right)}\left(I\right),{\hat{P}}_{X}^{\left(n\right)}\left(S\right)\right)$ that satisfies the equality in (Equation 11). This is essentially due to the fact that the equality in (Equation 11) forms a linear system of two equations with two variables, and thus, if it is consistent, it has either a unique solution or infinitely many solutions.

**Lemma** **1.**
*Consider the empirical output distribution ${\bar{P}}_{Y}^{\left(n\right)}$ in (3) obtained by a test described by the conditional probability distribtuion $P_{Y|X}$. Then, the following five statements are equivalent:*

*The system of equations in (Equation 11) has a unique solution;*

*The sensitivity $P_{Y|X}\left(I|I\right)$ and specificity $P_{Y|X}\left(S|S\right)$ satisfy*
(12a)$$P_{Y|X}\left(I|I\right)+P_{Y|X}\left(S|S\right)\neq 1;$$

*The sensitivity $P_{Y|X}\left(I|I\right)$ and the probability of a false positive $P_{Y|X}\left(I|S\right)$ satisfy*
(12b)$$P_{Y|X}\left(I|I\right)\neq P_{Y|X}\left(I|S\right);and$$

*The probability of a false negative $P_{Y|X}\left(S|I\right)$ and the specificity $P_{Y|X}\left(S|S\right)$ satisfy*
(12c)$$P_{Y|X}\left(S|S\right)\neq P_{Y|X}\left(S|I\right).$$

*The probability of a false positive $P_{Y|X}\left(I|S\right)$ and the probability of a false negative $P_{Y|X}\left(S|I\right)$ satisfy*
(12d)$$P_{Y|X}\left(I|S\right)+P_{Y|X}\left(S|I\right)\neq 1.$$



**Proof.** The proof of Lemma 1 follows from the fact that a unique solution to (Equation 11) is observed if and only if the determinant of the matrix
$$\left(\begin{array}{cc} P_{Y|X}\left(I|I\right) & P_{Y|X}\left(I|S\right)\\ P_{Y|X}\left(S|I\right) & P_{Y|X}\left(S|S\right) \end{array}\right)$$
is different than zero (Rouché–Fontené theorem [33]). That is,
(13)$$P_{Y|X}\left(I|I\right)P_{Y|X}\left(S|S\right)-P_{Y|X}\left(S|I\right)P_{Y|X}\left(I|S\right)\neq 0.$$The proof is complete by verifying that the expression in (13) is equivalent to those in (12). ☐

Note that all conditions in (12) are equivalent to each other, and thus, they are equivalent to the condition in (Equation 6).

The proof of Theorem 1 continues by showing that when such a unique solution exists, it is identical to the one shown in (8).

**Lemma** **2.**
*Consider a test $P_{Y|X}$ that satisfies at least one of the conditions in (12). Then, under the assumption that the empirical output distribution ${\bar{P}}_{Y}^{\left(n\right)}$ in (3) satisfies (Equation 7), the unique probability distribution ${\hat{P}}_{X}^{\left(n\right)}$ that satisfies (Equation 11) is:*
(14a)$$\begin{array}{ccc} {\hat{P}}_{X}^{\left(n\right)}\left(I\right) & = & \frac{1-{\bar{P}}_{Y}^{\left(n\right)}\left(I\right)-P_{Y|X}\left(S|S\right)}{1-P_{Y|X}\left(I|I\right)-P_{Y|X}\left(S|S\right)},and \end{array}$$
(14b)$$\begin{array}{ccc} {\hat{P}}_{X}^{\left(n\right)}\left(S\right) & = & \frac{{\bar{P}}_{Y}^{\left(n\right)}\left(I\right)-P_{Y|X}\left(I|I\right)}{1-P_{Y|X}\left(I|I\right)-P_{Y|X}\left(S|S\right)}. \end{array}$$


**Proof.** The proof of Lemma 2 follows from solving the system of equations in (Equation 11) and observing that ${\hat{P}}_{X}^{\left(n\right)}$ is a probability measure if and only if condition (Equation 7) holds. ☐

The rest of the proof of Theorem 1 consists of showing that the error vanishes with the number of test results. This is shown in three steps. The first step consists of showing that the total variation between $P_{X}$ and ${\hat{P}}_{X}^{\left(n\right)}$, denoted by ${∥P_{X}-{\hat{P}}_{X}^{\left(n\right)}∥}_{\mathrm{TV}}$, is equivalent to the total variation between $P_{Y}$ and ${\bar{P}}_{Y}^{\left(n\right)}$, denoted by ${∥P_{Y}-{\bar{P}}_{Y}^{\left(n\right)}∥}_{\mathrm{TV}}$, up to a scaling factor.

**Lemma** **3.**
*Consider a test $P_{Y|X}$ that satisfies at least one of the conditions in (12). Then, under the assumption that the empirical output distribution ${\bar{P}}_{Y}^{\left(n\right)}$ in (3) satisfies (Equation 7), the estimation ${\hat{P}}_{X}^{\left(n\right)}$ in (8) of $P_{X}$ satisfies*
(15)$${∥P_{X}-{\hat{P}}_{X}^{\left(n\right)}∥}_{\mathrm{TV}}=\frac{1}{|1-P_{Y|X}\left(I|I\right)-P_{Y|X}\left(S|S\right)|}{∥P_{Y}-{\bar{P}}_{Y}^{\left(n\right)}∥}_{\mathrm{TV}},$$

*where $P_{X}$ and $P_{Y}$ are the input and output probability distributions in (Equation 1) and (Equation 2), respectively.*


**Proof.** The proof of Lemma 3 follows from the definition of total variation in (Equation 4) and from equalities in (14). ☐

Note that Lemma 3 proves the intuition over which the proof of Theorem 1 is based on. That is, if ${\bar{P}}_{Y}^{\left(n\right)}$ is sufficiently close to $P_{Y}$, then ${\hat{P}}_{X}^{\left(n\right)}$ must be sufficiently close to $P_{X}$. The following lemma shows that the more test results are available, the closer ${\bar{P}}_{Y}^{\left(n\right)}$ and $P_{Y}$ are in total variation.

**Lemma** **4.**
*Consider a test $P_{Y|X}$ that satisfies at least one of the conditions in (12). Then, the empirical output distribution ${\bar{P}}_{Y}^{\left(n\right)}$ in (3) satisfies*
(16)$$\lim_{n\to \infty }{∥P_{Y}-{\bar{P}}_{Y}^{\left(n\right)}∥}_{\mathrm{TV}}=0,almost\;surely,$$

*where $P_{Y}$ is the ground-truth output probability distribution in (Equation 2).*


**Proof.** The proof of Lemma 4 is a consequence of the Theorem of Glivenko and Cantelli [34].

Finally, from Lemma 3 and Lemma 4, it holds that by increasing the number of tests, the error of approximating $P_{X}$ by ${\hat{P}}_{X}^{\left(n\right)}$ in (14) can be made arbitrarily small. The following lemma leverages this observation.

**Lemma** **5.**
*Consider a test $P_{Y|X}$ that satisfies at least one of the conditions in (12). Then, under the assumption that the empirical output distribution ${\bar{P}}_{Y}^{\left(n\right)}$ in (3) satisfies (Equation 7), the input distribution $P_{X}$ and the estimation ${\hat{P}}_{X}^{\left(n\right)}$ in (8) satisfy*
(17)$$\lim_{n\to \infty }{∥P_{X}-{\hat{P}}_{X}^{\left(n\right)}∥}_{\mathrm{TV}}=0,almost\;surerly.$$


**Proof.** The proof of Lemma 5 is an immediate consequence of both Lemma 3 and Lemma 4. ☐

This completes the proof of Theorem 1.

### 4.3. Connections to Maximum Likelihood Estimation

In this section, it is shown that the estimator presented in Theorem 1 is also the *maximum likelihood estimator*. For doing so, note that under the assumption that the prevalence ratio is ${\hat{P}}_{X}^{\left(n\right)}\left(I\right)\in \left[0,1\right]$, the probability of observing y∈{I,S}, as the result of testing any of the individuals of the population with a test described by the conditional probability distribution $P_{Y|X}$ is:(18)$${\hat{P}}_{Y}^{\left(n\right)}\left(y\right)≜\sum_{x\in \{I,S\}}P_{Y|X}\left(y|x\right){\hat{P}}_{X}^{\left(n\right)}\left(x\right).$$

From this perspective, the probability of observing the vector $y=\left(y_{1},y_{2},\ldots ,y_{n}\right)$, as the result of a testing campaign over a population of *n* individuals is
(19)$${\left({\hat{P}}_{Y}^{\left(n\right)}\left(I\right)\right)}^{n{\bar{P}}_{Y}^{\left(n\right)}\left(I\right)}{\left(1-{\hat{P}}_{Y}^{\left(n\right)}\left(I\right)\right)}^{n\left(1-{\bar{P}}_{Y}^{\left(n\right)}\left(I\right)\right)},$$
where ${\bar{P}}_{Y}^{\left(n\right)}$ and ${\hat{P}}_{Y}^{\left(n\right)}$ are defined in (3) and (18), respectively. Hence, the log-likelihood function $L:{\left\{I,S\right\}}^{n}\times \left[0,1\right]\to \left[0,1\right]$ is for all $y\in {\left\{I,S\right\}}^{n}$ and ${\hat{P}}_{X}^{\left(n\right)}\left(I\right)\in \left[0,1\right]$,
(20)$$\begin{array}{ccc} L(y,{\hat{P}}_{X}^{\left(n\right)}\left(I\right)) & = & n\left({\bar{P}}_{Y}^{\left(n\right)}\left(I\right)\ln\left({\hat{P}}_{Y}^{\left(n\right)}\left(I\right)\right)+\left(1-{\bar{P}}_{Y}^{\left(n\right)}\left(I\right)\right)\ln\left(1-{\hat{P}}_{Y}^{\left(n\right)}\left(I\right)\right)\right) \end{array}$$
(21)$$\begin{array}{cc} = & n({\bar{P}}_{Y}^{\left(n\right)}\left(I\right)\ln\left({\hat{P}}_{Y}^{\left(n\right)}\left(I\right)\right)-{\bar{P}}_{Y}^{\left(n\right)}\left(I\right)\ln\left({\bar{P}}_{Y}^{\left(n\right)}\left(I\right)\right) \end{array}$$
(22)$$\begin{array}{c} +\left(1-{\bar{P}}_{Y}^{\left(n\right)}\left(I\right)\right)\ln\left(1-{\hat{P}}_{Y}^{\left(n\right)}\left(I\right)\right)-\left(1-{\bar{P}}_{Y}^{\left(n\right)}\left(I\right)\right)\ln\left(1-{\bar{P}}_{Y}^{\left(n\right)}\left(I\right)\right)\; \end{array}$$
(23)$$\begin{array}{c} +{\bar{P}}_{Y}^{\left(n\right)}\left(I\right)\ln\left({\bar{P}}_{Y}^{\left(n\right)}\left(I\right)\right)+\left(1-{\bar{P}}_{Y}^{\left(n\right)}\left(I\right)\right)\ln\left(1-{\bar{P}}_{Y}^{\left(n\right)}\left(I\right)\right)) \end{array}$$
(24)$$\begin{array}{cc} = & -n\left(H\left({\bar{P}}_{Y}^{\left(n\right)}\right)+D\left({\bar{P}}_{Y}^{\left(n\right)}\left|\right|{\hat{P}}_{Y}^{\left(n\right)}\right)\right), \end{array}$$
where $H\left({\bar{P}}_{Y}^{\left(n\right)}\right)$ denotes the entropy of the probability distribution ${\bar{P}}_{Y}^{\left(n\right)}$; and $D\left({\bar{P}}_{Y}^{\left(n\right)}\left|\right|{\hat{P}}_{Y}^{\left(n\right)}\right)$ denotes the Kullback–Liebler divergence between the distributions ${\bar{P}}_{Y}^{\left(n\right)}$ and ${\hat{P}}_{Y}^{\left(n\right)}$. Given that $D\left({\bar{P}}_{Y}^{\left(n\right)}\left|\right|{\hat{P}}_{Y}^{\left(n\right)}\right)>0$, it follows that
(25)$$\begin{array}{ccc} L(y,{\hat{P}}_{X}^{\left(n\right)}\left(I\right)) & ⩽ & -nH\left({\bar{P}}_{Y}^{\left(n\right)}\right), \end{array}$$
where the equality holds if and only if $D\left({\bar{P}}_{Y}^{\left(n\right)}\left|\right|{\hat{P}}_{Y}^{\left(n\right)}\right)=0$. That is, when both ${\bar{P}}_{Y}^{\left(n\right)}$ and ${\hat{P}}_{Y}^{\left(n\right)}$ are identical. This observation leads to the conclusion that the log-likelihood function is maximized when the assumed prevalence ratio ${\hat{P}}_{X}^{\left(n\right)}\left(I\right)$ is such that ${\bar{P}}_{Y}^{\left(n\right)}$ in (3) and ${\hat{P}}_{Y}^{\left(n\right)}$ in (18) are identical, which is induces the system of equations in (Equation 11) and in which the unique solution is formed by the equalities in (8). This proves that the estimator in Theorem (1) is the unique maximum likelihood estimator.

## 5. Final Remarks

This section highlights some of the conclusions drawn from Lemma 1–5 using a numerical analysis in particular examples. In the following examples, the data is artificially generated. That is, for a given prevalence ratio $P_{X}\left(I\right)$, an *n*-dimensional vector $x=(x_{1}$, $x_{2}$, *…*, $x_{n})\in {\left\{I,S\right\}}^{n}$ is generated such that for all t∈{1,2,…,n}, $x_{t}$ is a realization of a random variable $X∼P_{X}$ and represents the state of individual *t*. Given a test $P_{Y|X}$, an *n*-dimensional vector $y=(y_{1}$, $y_{2}$, *…*, $y_{n})\in {\left\{I,S\right\}}^{n}$ is generated such that for all t∈{1,2,…,n}, $y_{t}$ is the realization of a random variable $Y_{t}∼P_{Y|X=x_{t}}$ and represents the result of the test of individual *t*. Using the vector y, the fraction of positive tests ${\bar{P}}_{Y}^{\left(n\right)}\left(I\right)$ is calculated using (3); and the estimation ${\hat{P}}_{X}^{\left(n\right)}\left(I\right)$ of the prevalence ratio $P_{X}\left(I\right)$ is calculated using ([Disp-formula FD8a-ijerph-17-05328]). Figure 2 shows this procedure.

From this perspective, the analysis is based on simulated testing campaigns. Note that the use of simulated data allows knowing the actual prevalence ratio, which enables analyzing the estimation error. This is rarely possible with data from actual testing campaigns.

**Example** **1.**
*Consider a population of n=10,000 individuals with prevalence $P_{X}\left(I\right)=0.4$. Assume that all individuals are tested with identical tests $P_{Y|X}$.*


**Example** **2.**
*Consider a population of n=100,000 individuals with prevalence $P_{X}\left(I\right)=0.4$. Assume that all individuals are tested with identical tests $P_{Y|X}$.*


**Example** **3.**
*Consider a population of n=100,000,000 individuals with prevalence $P_{X}\left(I\right)=0.4$. Assume that all individuals are tested with identical tests $P_{Y|X}$.*


In Figure 3, Figure 4, Figure 5, Figure 6, Figure 7 and Figure 8, the actual prevalence ratio $P_{X}\left(I\right)$ is plotted with a straight black line; the estimation ${\hat{P}}_{X}^{\left(n\right)}$ of $P_{X}$ is plotted with red circles; the fraction of positive tests ${\bar{P}}_{Y}^{\left(n\right)}\left(I\right)$ is plotted with blue diamonds; and the value of $P_{Y}\left(I\right)$ in (Equation 2) is plotted with a dashed red line. In Figure 3, Figure 5 and Figure 7, these values are plotted as a function of the specificity $P_{Y|X}\left(S|S\right)$ for a fixed sensitivity. Alternatively, in Figure 4, Figure 6 and Figure 8, these values are plotted as a function of the sensitivity $P_{Y|X}\left(I|I\right)$ for a fixed specificity. For each of the examples, one vector $x\in {\left\{I,S\right\}}^{n}$ is generated. In all figures, Figure 3, Figure 4, Figure 5, Figure 6, Figure 7 and Figure 8, each plotted point of ${\bar{P}}_{Y}^{\left(n\right)}\left(I\right)$ (blue diamonds) and ${\hat{P}}_{X}^{\left(n\right)}\left(I\right)$ (red circles) is calculated using a single vector y generated by the same vector x, according to the corresponding values of sensitivity $P_{Y|X}\left(I|I\right)$ and specificity $P_{Y|X}\left(S|S\right)$, as described above. In the following sections, some remarks based on these examples are presented.

### 5.1. Relevance of the Sensitivity and Specificity

One of the main observations to be highlighted from this numerical analysis is that there exists an important difference between the fraction of positive tests ${\bar{P}}_{Y}^{\left(n\right)}\left(I\right)$ and the actual prevalence ratio $P_{X}\left(I\right)$ due to the sensitivity and specificity of the tests. This difference is clearly depicted in Figure 3, Figure 4, Figure 5, Figure 6, Figure 7 and Figure 8, which together with the mathematical analysis presented before, highlights the conclusion that the fraction of positive tests should not be used as an estimation of the prevalence ratio in public health policy-making.

The following lemma determines the influence of the sensitivity and specificity on ${\bar{P}}_{Y}^{\left(n\right)}\left(I\right)$. For doing so, note that from Lemma (2), it holds that the fraction of individuals reporting positive tests ${\bar{P}}_{Y}^{\left(n\right)}\left(I\right)$ satisfies:(26)$${\bar{P}}_{Y}^{\left(n\right)}\left(I\right)=1-P_{Y|X}\left(S|S\right)\left(1-{\hat{P}}_{X}^{\left(n\right)}\left(I\right)\right)-\left(1-{\hat{P}}_{Y|X}\left(I|I\right)\right){\hat{P}}_{X}^{\left(n\right)}\left(I\right).$$

**Lemma** **6.**
*Consider a test $P_{Y|X}$ that satisfies at least one of the conditions in (12). Then, given the empirical output distribution ${\bar{P}}_{Y}^{\left(n\right)}$ in (3) and assuming that it satisfies (Equation 7), the following statements hold:*


*The fraction ${\bar{P}}_{Y}^{\left(n\right)}\left(I\right)$ of positive tests linearly decreases with the specificity of the test $P_{Y|X}\left(S|S\right)$;*

*The fraction ${\bar{P}}_{Y}^{\left(n\right)}\left(I\right)$ of positive tests linearly increases with the probability of a false positive of the test $P_{Y|X}\left(I|S\right)$;*

*The fraction ${\bar{P}}_{Y}^{\left(n\right)}\left(I\right)$ of positive tests linearly increases with the sensitivity of the test $P_{Y|X}\left(I|I\right)$; and*

*The fraction ${\bar{P}}_{Y}^{\left(n\right)}\left(I\right)$ of positive tests linearly decreases with the probability of a false negative of the test $P_{Y|X}\left(S|I\right)$.*



**Proof.** The proof of Lemma 6 consists in verifying that the derivative of ${\bar{P}}_{Y}^{\left(n\right)}$ in (26) with respect to $P_{Y|X}\left(S|S\right)$ is negative; with respect to $P_{Y|X}\left(I|S\right)$ is positive; with respect to $P_{Y|X}\left(I|I\right)$ is positive; and with respect to $P_{Y|X}\left(S|I\right)$ is negative.

The statements in Lemma 6 become evident in Figure 3, Figure 5 and Figure 7. In these figures, it is shown that the fraction of positive tests increases with the sensitivity; where as in Figure 4, Figure 6 and Figure 8, it is shown that the fraction of positive tests decreases with the specificity, c.f., Lemma 6. From this perspective, tests might lead to countings in which the fraction of individuals reporting positive testing results ${\bar{P}}_{Y}^{\left(n\right)}\left(I\right)$ is bigger than the actual prevalence ratio $P_{X}\left(I\right)$, i.e., ${\bar{P}}_{Y}^{\left(n\right)}\left(I\right)>P_{X}\left(I\right)$. Alternatively, tests might also lead to estimations in which the fraction of individuals reporting positive testing results ${\bar{P}}_{Y}^{\left(n\right)}\left(I\right)$ is smaller than the actual prevalence ratio $P_{X}\left(I\right)$, i.e., ${\bar{P}}_{Y}^{\left(n\right)}\left(I\right)<P_{X}\left(I\right)$. These observations highlight the relevance of using the estimation ${\hat{P}}_{X}^{\left(n\right)}$ of $P_{X}$ for decision and policy making instead of ${\bar{P}}_{Y}^{\left(n\right)}$, which includes false positives and false negatives.

### 5.2. Tests whose Results are Useless

In Figure 3, Figure 4, Figure 5, Figure 6, Figure 7 and Figure 8, the value of the sensitivity $P_{Y|X}\left(I|I\right)$ and specificity $P_{Y|X}\left(S|S\right)$ that satisfy $P_{Y|X}\left(I|I\right)+P_{Y|X}\left(S|S\right)=1$ are plotted with a blue dash-dot vertical line. Note that for these specific values of sensitivity and specificity, the estimation ${\hat{P}}_{X}^{\left(n\right)}\left(I\right)$ of $P_{X}\left(I\right)$ is not plotted. The following lemmas shed some light into this singularity.

**Lemma** **7.**
*Consider the empirical output distribution ${\bar{P}}_{Y}^{\left(n\right)}$ in (3) obtained by a test described by the conditional probability distribtuion $P_{Y|X}$. Then, the following five statements are equivalent:*

*The system of equations in (Equation 11) has infinitely many solutions;*

*The sensitivity $P_{Y|X}\left(I|I\right)$ and specificity $P_{Y|X}\left(S|S\right)$ satisfy*
(27a)$$P_{Y|X}\left(I|I\right)+P_{Y|X}\left(S|S\right)=1;$$

*The sensitivity $P_{Y|X}\left(I|I\right)$ and the probability of a false positive $P_{Y|X}\left(I|S\right)$ satisfy*
(27b)$$P_{Y|X}\left(I|I\right)=P_{Y|X}\left(I|S\right);$$

*The probability of a false negative $P_{Y|X}\left(S|I\right)$ and the specificity $P_{Y|X}\left(S|S\right)$ satisfy*
(27c)$$P_{Y|X}\left(S|S\right)=P_{Y|X}\left(S|I\right);and$$

*The probability of a false positive $P_{Y|X}\left(I|S\right)$ and the probability of a false negative $P_{Y|X}\left(S|I\right)$ satisfy*
(27d)$$P_{Y|X}\left(I|S\right)+P_{Y|X}\left(S|I\right)=1.$$



**Proof.** The proof of Lemma 7 follows from the theorem of Rouché and Fontené [33] that states that when the system in (Equation 11) is consistent, it has infinitely many solutions if the determinant of the matrix
$$\left(\begin{array}{cc} P_{Y|X}\left(I|I\right) & P_{Y|X}\left(I|S\right)\\ P_{Y|X}\left(S|I\right) & P_{Y|X}\left(S|S\right) \end{array}\right)$$
is not full rank. When such a matrix is not full rank, its determinant is zero. That is,
(28)$$P_{Y|X}\left(I|I\right)P_{Y|X}\left(S|S\right)-P_{Y|X}\left(S|I\right)P_{Y|X}\left(I|S\right)=0.$$The proof is completed by verifying that the expression in (28) is equivalent to those in (27). ☐

When at least one of the equalities in (27) is satisfied, nothing meaningful can be said about $P_{X}$ based on the data. This is essentially because any probability distribution ${\hat{P}}_{X}^{\left(n\right)}$ satisfies the equality in (Equation 11). The following lemma reinforces this statement in terms of information measures.

**Lemma** **8.**
*Consider a test $P_{Y|X}$ that satisfies at least one of the conditions in (27). Hence, the following statements are equivalent:*

*Given the output empirical distribution ${\bar{P}}_{Y}^{\left(n\right)}$ obtained from the data y as in (3), any probability distribution ${\hat{P}}_{X}^{\left(n\right)}$ on {I,S} satisfies the equality in (Equation 11);*

*Two random variables X and Y, in which the joint probability distribution $P_{XY}$ satisfies (Equation 1), have zero mutual information; and*

*Two random variables X and Y, in which the joint probability distribution $P_{XY}$ satisfies (Equation 1), are independent.*



**Proof.** The first statement is a consequence of Lemma 7; the second statement follows from the fact that under any of the assumptions in (27), the mutual information satisfies
(29)$$\begin{array}{ccc} I(X;Y) & ≜ & \sum_{x\in \{I,S\}}\sum_{y\in \{I,S\}}P_{X}\left(x\right)P_{Y|X}\left(y|x\right)\log_{2}\left(\frac{P_{Y|X}\left(y|x\right)}{P_{Y}\left(y\right)}\right) \end{array}$$
(30)$$\begin{array}{cc} = & 0, \end{array}$$
where $P_{X}$, $P_{Y|X}$, and $P_{Y}$ satisfy the equality in (Equation 1). The third statement follows from the fact that two random variables are independent if and only if their mutual information is zero. ☐

Lemma 8 shows that when at least one of the conditions in (27) holds, the output probability distribution $P_{Y}$ does not provide any information about the input probability distribution $P_{X}$. That is, nothing can be said about $P_{X}$ based on the data y.

Despite the singularity, the values of specificity and sensitivity in which the sum is close to one, i.e., around the singularity, are also worthy of discussion. Note that for some 1>ϵ>0, the absolute difference $|P_{X}\left(I\right)-{\hat{P}}_{X}^{\left(n\right)}\left(I\right)|$ is bigger when the sensibility and specificity satisfy $|1-P_{Y|X}\left(S|S\right)-P_{Y|X}\left(I|I\right)|<\epsilon$ than when these parameters satisfy $|1-P_{Y|X}\left(S|S\right)-P_{Y|X}\left(I|I\right)|>\epsilon$. These observations are justified by the fact that the total variation ${∥P_{X}-{\hat{P}}_{X}^{\left(n\right)}∥}_{\mathrm{TV}}$ is equal to ${∥P_{Y}-{\bar{P}}_{Y}^{\left(n\right)}∥}_{\mathrm{TV}}$ up to a constant factor, as shown in Lemma 3. Such a factor is indeed $\frac{1}{|1-P_{Y|X}\left(I|I\right)-P_{Y|X}\left(S|S\right)|}$, and thus, larger errors are expected around the singularity for the same finite numbers of tests *n*. This is evident in the numerical analysis. In Example 1, i.e., Figure 3 and Figure 4, around the singularity, the estimations ${\hat{P}}_{X}^{\left(n\right)}$ of $P_{X}$ appear more disperse than the estimations in Example 3, i.e., Figure 7 and Figure 8.

### 5.3. Impact of the Number of Tests.

Figure 3, Figure 4, Figure 5, Figure 6, Figure 7 and Figure 8 show that when the parameters of the test satisfy at least one of the conditions in (12) and there exist a sufficiently large number of test results, it is always possible to obtain an estimation ${\hat{P}}_{X}^{\left(n\right)}\left(I\right)$ of the prevalence ratio $P_{X}\left(I\right)$. This is independent of the exact values of the specificity and sensitivity as long as (12) holds. More importantly, the reliability of such estimation increases with the number of test results. For instance, compare the estimations in Examples 1 and 3. The implications of this observation are very important in practical terms. This shows that if the objective of a testing campaign against SARS-CoV-2 is to determine the prevalence ratio, the quality of the tests is not important. This is essentially because testing with low quality tests (low sensitivity and low specificity) or high quality tests (high sensitivity and high specificity) leads to identical results in terms of the estimation error, when a large number of tests is performed. Nonetheless, when a low number of tests is available, it is worth noting that when the sensitivity $P_{Y|X}\left(I|I\right)$ and specificity $P_{Y|X}\left(S|S\right)$ satisfy $|1-P_{Y|X}\left(S|S\right)-P_{Y|X}\left(I|I\right)|>1-\epsilon$, for some 0<ϵ<1, the smaller ϵ, the smaller the estimation error of the prevalence ratio, c.f., Lemma 3. This observation is of paramount importance as it implies that smaller estimation errors are observed when the sum of the sensitivity and specificity is bounded away from one. This said, the key parameter for reducing the estimation error is the number of tests.

## 6. Conclusions

In this work, it has been shown that estimating the prevalence ratio of a condition, for example, a SARS-Cov-2 infection, by the ratio between the number of positive test results and the total number of tests leads to excessive estimation errors when tests are unreliable. This is simply due to the fact that unreliable tests, i.e., tests in which probabilities of false positives and false negatives are nonzero, lead to some individuals exhibiting the condition to observe negative test results (false negatives), and some individuals who do not exhibit the condition to observe positive results (false positives). From this perspective, an estimation of the prevalence ratio using data obtained from tests must take into account both the sensitivity and the specificity of the tests. Theorem 1 provides an estimation of the prevalence ratio with an estimation error that decreases with the number of tests.

Another important conclusion of this work is that testing campaigns using tests for which the sum of the sensitivity and specificity is different than one, always allow a reliable estimation of the prevalence ratio (Lemma 1 in Section 4) subject to a sufficiently large number of individuals being tested. Alternatively, testing campaigns using tests for which the sum of the sensitivity and the specificity is equal to one, lead to data from which it is impossible to estimate the prevalence ratio even with infinitely many tests (Lemma 7 in Section 4).

A final conclusion is that for estimating the prevalence ratio of a given condition, i.e., a SARS-CoV-2 infection, the key parameter for reducing the estimation error is the number of tests. Surprisingly, as long as the sum of the sensitivity and specificity of the tests is different than one, the exact values of both sensitivity and specificity have very little impact in the estimation when the number of tests is sufficiently large.

## Figures and Tables

**Figure 1 ijerph-17-05328-f001:**
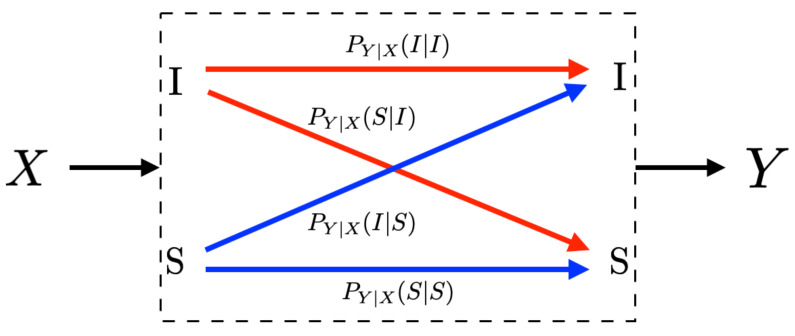
A SARS-CoV-2 test represented by a random transformation from {I,S} into {I,S} via the conditional probability distribution $P_{Y|X}$.

**Figure 2 ijerph-17-05328-f002:**
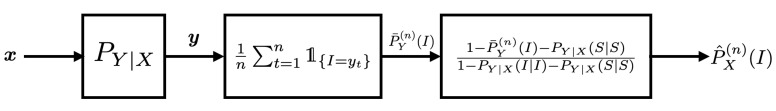
Relation between the input vector $x=\left(x_{1},x_{2},\ldots ,x_{n}\right)\in {\left\{I,S\right\}}^{n}$ (state of the individuals); the output vector $y=\left(y_{1},y_{2},\ldots ,y_{n}\right)\in {\left\{I,S\right\}}^{n}$ (result of the tests); the calculation of the fraction of positive tests ${\bar{P}}_{Y}^{\left(n\right)}\left(I\right)$ in (3); and estimation of the prevalence ratio ${\hat{P}}_{X}^{\left(n\right)}\left(I\right)$ in ([Disp-formula FD8a-ijerph-17-05328]).

**Figure 3 ijerph-17-05328-f003:**
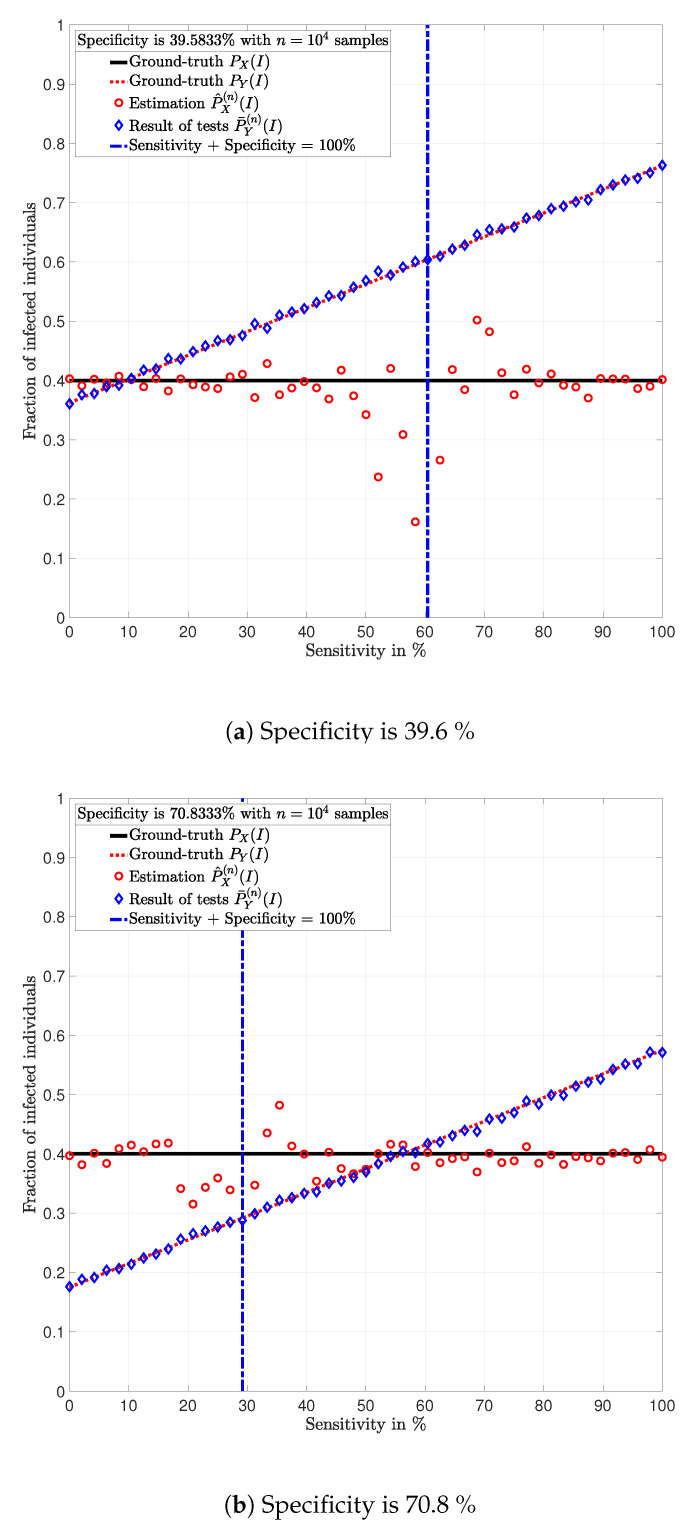
Population in which the fraction of individuals infected with SARS-CoV-2 is *P_X_* = 0.4 and *n* = 10, 000 individuals are tested (Example 1).

**Figure 4 ijerph-17-05328-f004:**
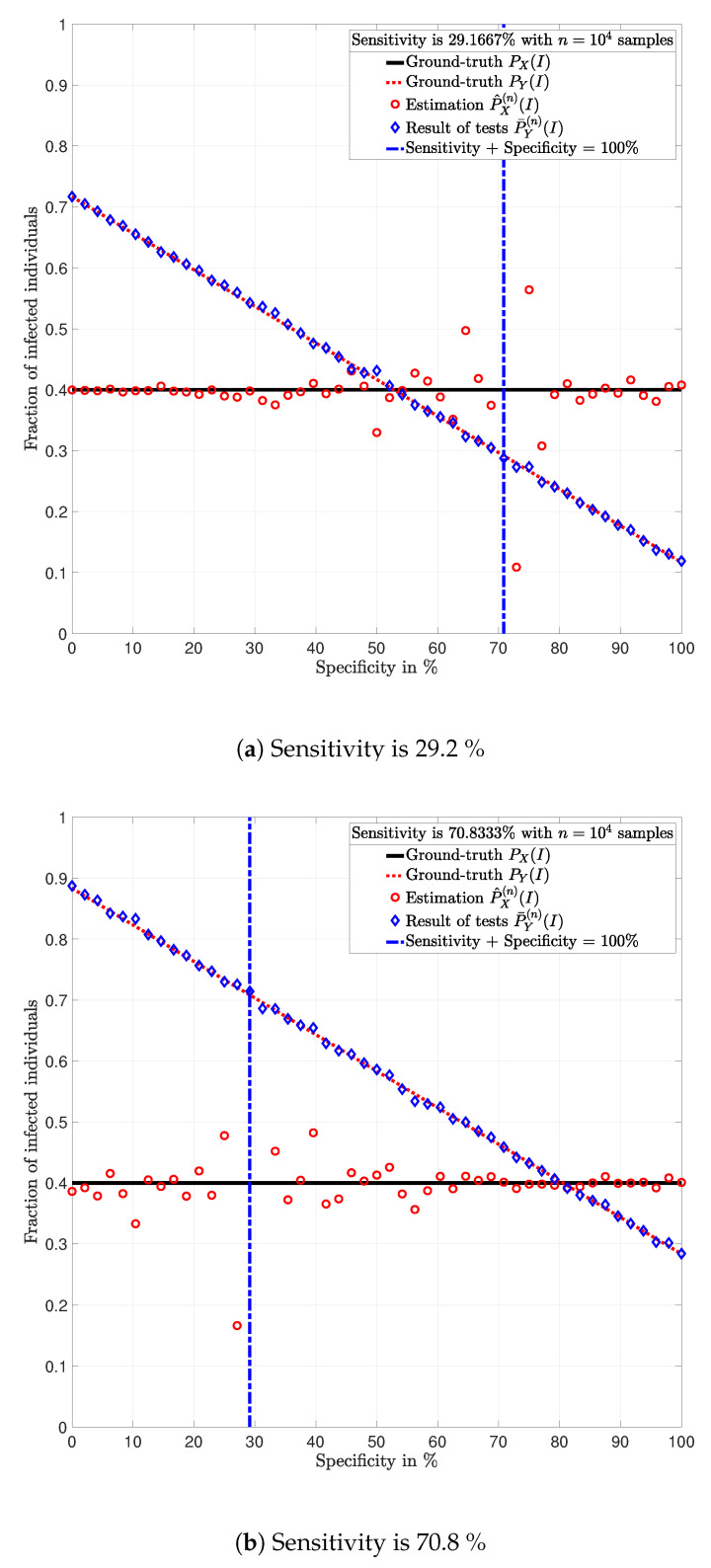
Population in which the fraction of individuals infected with SARS-CoV-2 is *P_X_* = 0.4 and *n* = 10, 000 individuals are tested (Example 1).

**Figure 5 ijerph-17-05328-f005:**
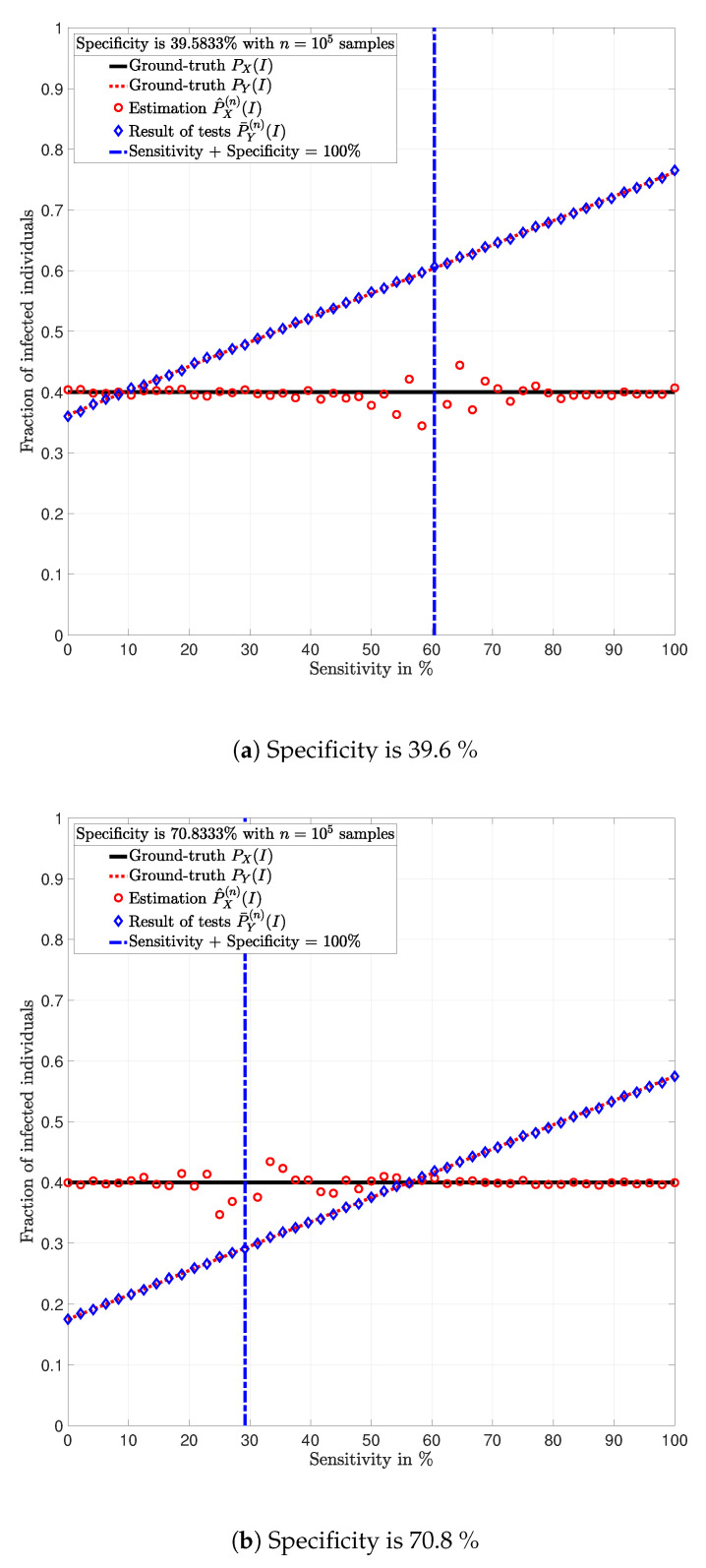
Population in which the fraction of individuals infected with SARS-CoV-2 is *P_X_* = 0.4 and *n* = 100, 000 individuals are tested (Example 2).

**Figure 6 ijerph-17-05328-f006:**
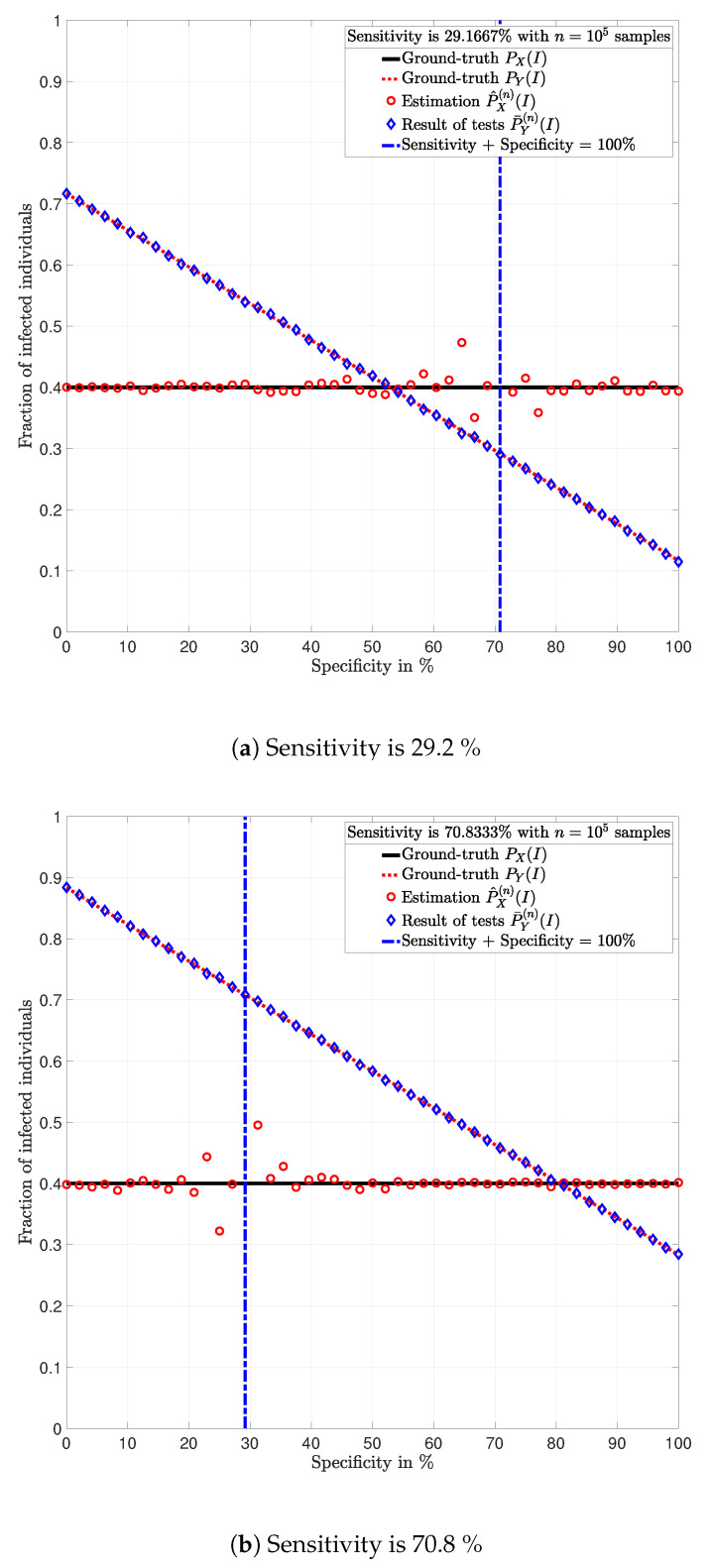
Population in which the fraction of individuals infected with SARS-CoV-2 is *P_X_* = 0.4 and *n* = 100, 000 individuals are tested (Example 2).

**Figure 7 ijerph-17-05328-f007:**
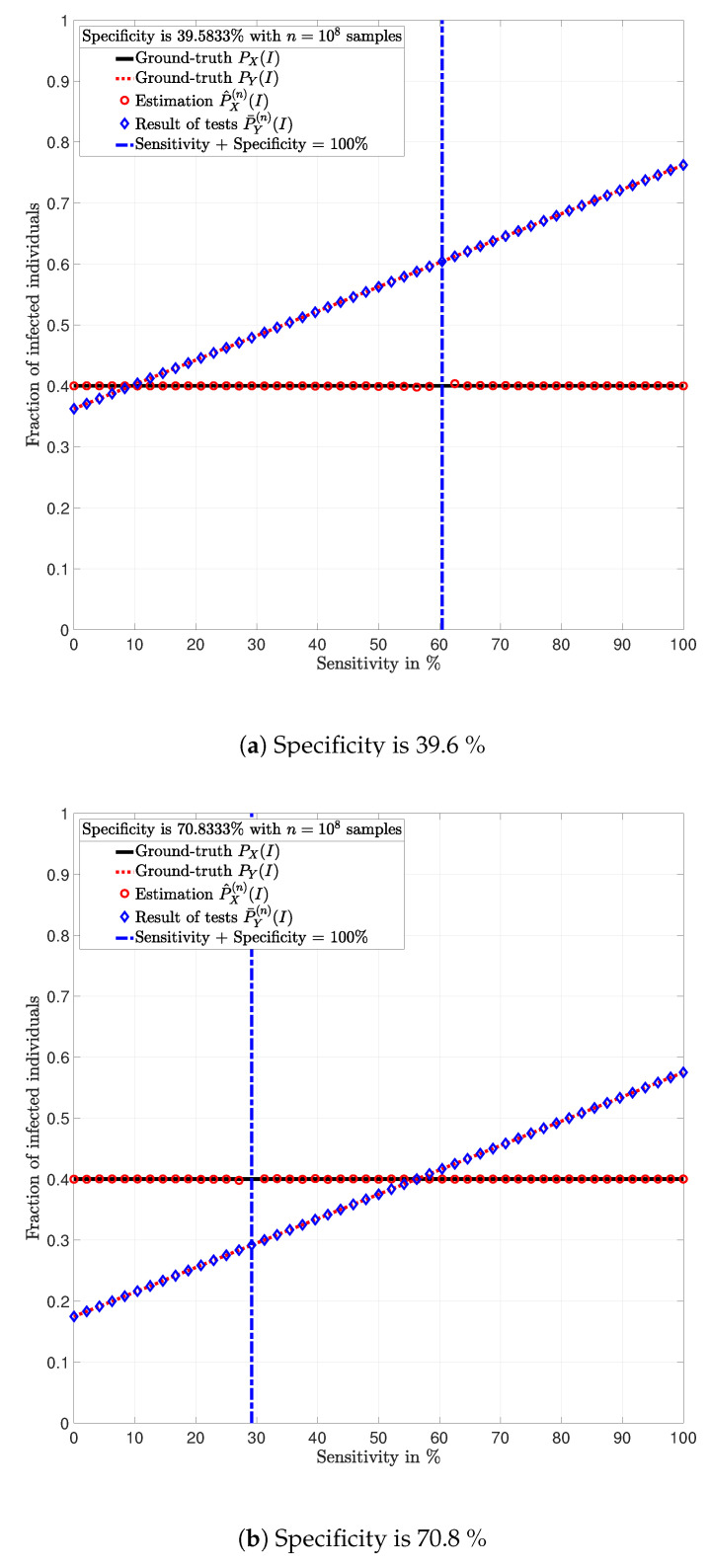
Population in which the fraction of individuals infected with SARS-CoV-2 is *P_X_* = 0.4 and *n* = 100, 000, 000 individuals are tested (Example 3).

**Figure 8 ijerph-17-05328-f008:**
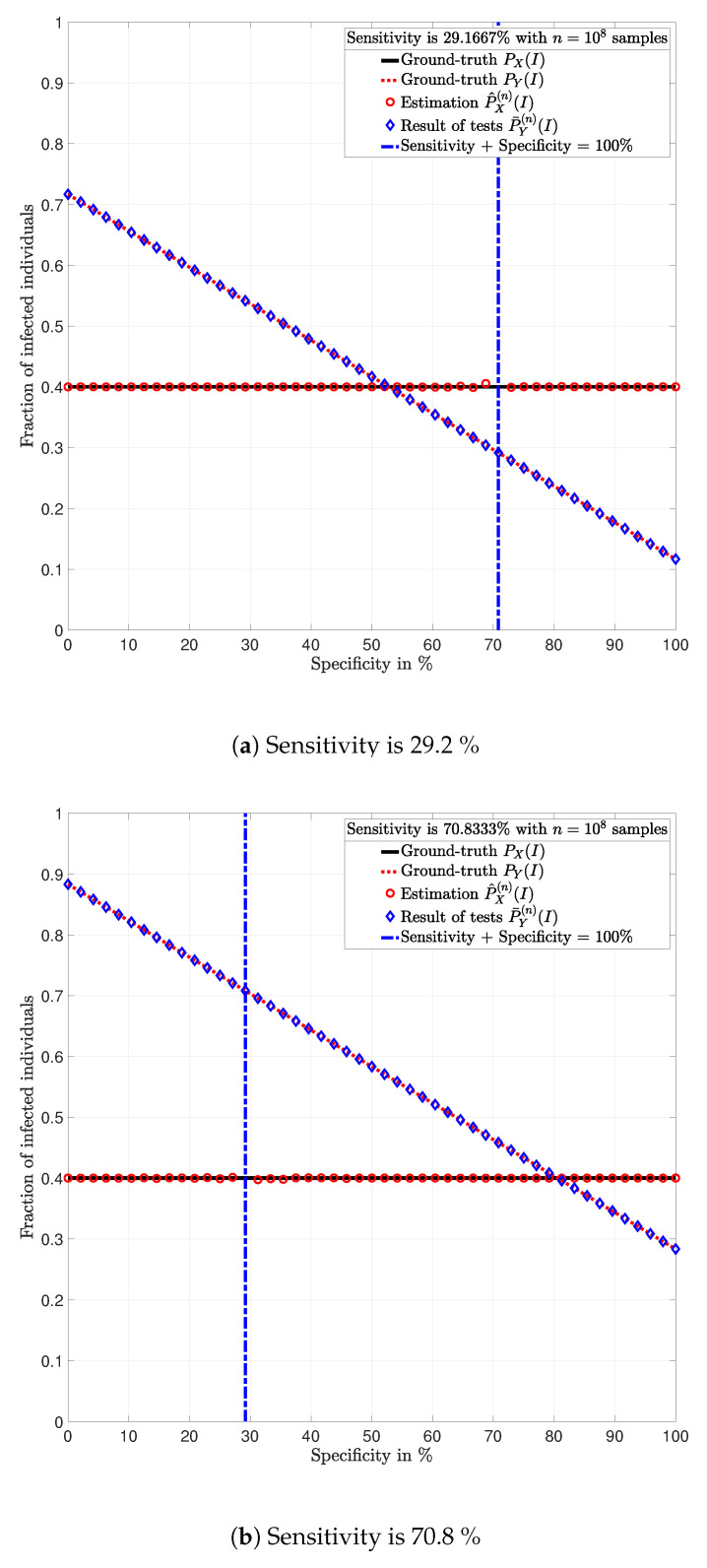
Population in which the fraction of individuals infected with SARS-CoV-2 is *P_X_* = 0.4 and *n* = 100, 000, 000 individuals are tested (Example 3).

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
