# Peer review of "On the True Number of COVID-19 Infections: Effect of Sensitivity, Specificity and Number of Tests on Prevalence Ratio Estimation"

_ijerph, 2020, doi:10.3390/ijerph17155328_

Round 1

Reviewer 1 Report

Dear Authors,

Thank you for your interesting proposal. However, this manuscript is interesting, please follow the reviewer's comments:

  1. The contributions need rewrite more clear.
  2. To describe your proposed algorithm, please add a block-diagram about your proposed algorithm in the introduction and explain it step-by-step.
  3. How you can proof that, the relation between infections and positive test is reliable and how you can proof the stability.
  4. You estimate the number of infections to find the estimation error. How about prediction?
  5. If we increase the parameters, the accuracy of test and detection are increasing?
  6. To reduce the estimation error, which techniques can be recommended?
  7. What is the advantages of proposed method compare to machine-learning based algorithm. 

Author Response

Dear Reviewer, 

Thank you for your time on reviewing this.
In the attached document you will find the answers to each of the comments you suggested during your revision. 

Best wishes,

Samir
On Behalf of all authors

Reviewer 2 Report

I suggest to improve literature analysis. In this form it's no evident the originality of the paper as compared to the existent literature. In addition it's necessary to improve conclusions. I have made this suggestions to improve literature review in which the authors could highlight what was analysed in the past. In this way the originality of the paper could be more evident. Therefore, I suggest to improve conclusion and putting in evidence the new apport respect to the past researches. Covid 19 is new, but Covid is an old virus and their is a vast literature that measures it.

Author Response

(The authors gave the same response as above.)

Reviewer 3 Report

Dear authors

Greetings

Nice work! congrats. I send you some suggestions (attached doc).

Success and best regards

Author Response

(The authors gave the same response as above.)

Round 2

Reviewer 1 Report

Dear Authors,

Thank you for your cover letter and revised manuscript, as well. Regarding the second round review, the reviewer believes that, this manuscript needs minor revision regarding the following comment.

  • However, the authors believe that they didn't use algorithm in this manuscript and they only use formulation!!!, the reviewer believes that, these formulations are extracted from methodology. So, to describe your proposed method, please add a block-diagram about your proposed
    technique in the introduction and explain sub-blocks step-by-step.

Regards,

Author Response

Dear Reviewers,

Thank you very much for your comments and suggestions. We have prepared a new version of the paper in which all your suggestions have been addressed. 

Best wishes,

Samir
